

# Microbe-ID: an open source toolbox for microbial genotyping and species identification

Javier F. Tabima[1], Sydney E. Everhart[1,5], Meredith M. Larsen[2], Alexandra J. Weisberg[1], Zhian N. Kamvar[1], Matthew A. Tancos[3], Christine D. Smart[3], Jeff H. Chang[1,4] and Niklaus J. Grünwald[1,2,3,4]

[1] Department of Botany and Plant Pathology, Oregon State University, Corvallis, OR, United States
[2] Horticultural Crops Research Laboratory, USDA Agricultural Research Service, Corvallis, OR, United States
[3] Plant Pathology and Plant-Microbe Biology Section, School of Integrative Plant Science, Cornell University, Geneva, NY, United States
[4] Molecular and Cellular Biology Graduate Program and Center for Genome Biology and Biocomputing, Oregon State University, Corvallis, OR, United States
[5] Current affiliation: Department of Plant Pathology, University of Nebraska, Lincoln, NE, United States

Corresponding author
Niklaus J. Grünwald,
nik.grunwald@ars.usda.gov,
grunwaln@science.oregonstate.edu

## ABSTRACT

Development of tools to identify species, genotypes, or novel strains of invasive organisms is critical for monitoring emergence and implementing rapid response measures. Molecular markers, although critical to identifying species or genotypes, require bioinformatic tools for analysis. However, user-friendly analytical tools for fast identification are not readily available. To address this need, we created a web-based set of applications called Microbe-ID that allow for customizing a toolbox for rapid species identification and strain genotyping using any genetic markers of choice. Two components of Microbe-ID, named Sequence-ID and Genotype-ID, implement species and genotype identification, respectively. Sequence-ID allows identification of species by using BLAST to query sequences for any locus of interest against a custom reference sequence database. Genotype-ID allows placement of an unknown multilocus marker in either a minimum spanning network or dendrogram with bootstrap support from a user-created reference database. Microbe-ID can be used for identification of any organism based on nucleotide sequences or any molecular marker type and several examples are provided. We created a public website for demonstration purposes called Microbe-ID (microbe-id.org) and provided a working implementation for the genus *Phytophthora* (phytophthora-id.org). In *Phytophthora*-ID, the Sequence-ID application allows identification based on ITS or *cox* spacer sequences. Genotype-ID groups individuals into clonal lineages based on simple sequence repeat (SSR) markers for the two invasive plant pathogen species *P. infestans* and *P. ramorum*. All code is open source and available on github and CRAN. Instructions for installation and use are provided at https://github.com/grunwaldlab/Microbe-ID.

## BACKGROUND

Development of tools for identification of species, genotypes or strains is critical for monitoring emergence of invasive organisms such as *Phytophthora ramorum* causing

sudden oak death (*Grünwald, Goss & Press, 2008*), *Hymenoscyphus fraxineus* causing ash dieback (*Gross et al., 2014*), *Aphanomyces astaci* causing crayfish plague (*Holdich et al., 2009*), *Cryptococcus gattii* causing cryptococcosis and meningitis (*Byrnes III et al., 2010*), or Methicillin-resistant *Staphylococcus aureus* causing invasive MRSA disease (*Klevens et al., 2007*). Molecular markers provide a rapid means for identification, but require various bioinformatics tools for identification of species and/or novel genotypes. In eukaryotes, sequences from the rRNA internal transcribed spacer (ITS) region and various mitochondrial DNA regions are used to separate discrete species (*Coleman, 2003*; *Coleman, 2007*). ITS and mtDNA markers are now the most widely used markers in plants (*Coleman, 2007*; *Coleman, 2009*), fungi (*James et al., 2006*), corals (*Grajales, Aguilar & Sánchez, 2007*), and oomycetes (*Cooke et al., 2012*; *Robideau et al., 2011*) and have been coined "DNA barcodes" because of their broad ability to distinguish species (*Schoch et al., 2012*). Classification of individuals using various molecular markers has recently increased. Multi-locus sequence types (MLST) are being widely used by researchers working with bacterial taxa to reveal the identity of samples by classification relative to known reference strains (*Maiden et al., 2013*). Other molecular markers or methods used to distinguish genotypes might include microsatellites (or simple sequence repeats) to identify strains and clonal lineages (*Cooke et al., 2012*; *Ivors et al., 2006*), DNA sequences for specific genic regions (*Maiden et al., 2013*), single nucleotide polymorphism (SNP) genotyping using reduced representation approaches (*Grünwald, McDonald & Milgroom, in press*) such as RAD-seq (*Etter et al., 2010*) or genotyping by sequencing (GBS, *Elshire et al., 2011*), or genome wide SNP genotyping (*Huang et al., 2009*).

In addition to the molecular methods developed, different types of online databases have been implemented to identify species using these molecular methods within groups of organisms. Examples of these databases are FungiDB for fungi and fungal-like organisms (*Stajich et al., 2011*), EuPathDB for eukaryotic organisms (*Aurrecoechea et al., 2013*), and the *Phytophthora* database which allows entries by experts in the *Phytophthora* community from different labs or countries for different species of the genus (*Park et al., 2008*). We previously reported on our development of a database for *Phytophthora* species and genotype identification using web tools to identify species using common barcodes, enabling the conjunction of modern laboratory techniques with highly curated databases for species identification (*Grünwald et al., 2011*).

Our objective here was to report the development of a toolbox for microbe identification (Microbe-ID) that can readily be customized for sequence based species identification (Sequence-ID) or molecular marker-based identification of genotypes (Genotype-ID) for any group of organisms. Our objectives were two-fold: (1) to implement Microbe-ID as a demonstration site that is customizable for any group of organisms and (2) to demonstrate a working implementation at *Phytophthora*-ID.org version 2.0 with significant updates from version 1.0 (*Grünwald et al., 2011*). Microbe-ID includes two modules, Sequence-ID and Microbe-ID. Sequence analysis is implemented based on use of a well characterized barcode region for the genus *Phytophthora*, but can be implemented to use any barcode sequence of interest. Genotype analysis can be implemented to use a variety of marker data types. To demonstrate the breadth of the developed tools and applicability to the

diversity of microorganisms, the following three examples were included: codominant microsatellite data (SSR/Microsatellite) for the oomycete *P. ramorum* (*Grünwald et al., 2009*), concatenated Multi Locus Sequence Type (MLST) or individual locus sequences for the bacterium *Clavibacter michiganensis* subsp. *michiganensis* (*Tancos, Lange & Smart, 2015*, Fig. S3), and dominant Amplified Fragment Length Polymorphism (Binary (AFLP) data, Fig. S4) for the oomycete *Aphanomyces euteiches* (*Grünwald & Hoheisel, 2006*). Moreover, Genotype-ID can be expanded to include other marker systems including gene sequences for resistance to antibiotics or fungicides as well as presence/absence polymorphisms for effector genes or other adaptive loci. Two sequence databases were developed that help us demonstrate the utility of Microbe-ID. These databases are for the genus *Phytophthora* (*Phytophthora*-ID) with two sequence databases containing over 110 species that are mostly plant pathogens (*Kroon et al., 2012*), and two genotyping databases for populations of the potato late blight pathogen, *P. infestans*, and the sudden oak death pathogen, *P. ramorum* (Genotype-ID) (*Grünwald, Goss & Press, 2008*; *Kamoun et al., 2015*). All of these tools are readily customizable and open source (http://www.github.com/grunwaldlab/microbe-ID), provided as a demonstration site (http://microbe-id.org/), and a working implementation of Sequence-ID and Genotype-ID that we use in our own work for the genus *Phytophthora* (phytophtora-id.org). Finally, a companion paper describes application of Microbe-ID for the implementation of a new website, Gall-ID, with novel tools for identification of gall-forming bacteria (*Davis II et al., 2016*).

## The Microbe-ID toolbox

We developed a template website named Microbe-ID (http://microbe-id.org/) with two separate modules, Sequence-ID and Genotype-ID, for sequence-based species identification and genotyping, respectively. The website is written using bootstrap (http://getbootstrap.com/), a HTML, CSS, and JS framework for developing responsive, mobile first projects on the web. Specific instructions, code, and resources necessary for implementation of Microbe-ID are provided on the GitHub repository (http://www.github.com/grunwaldlab/microbe-ID). The server currently hosting Microbe-ID is running Centos Linux release 6.6, NCBI BLAST 2.2.28+ (*Altschul et al., 1990*), MAFFT version 7.221 (*Katoh et al., 2002*), and R version 3.1.2. Below we describe specific tools required for customization of each module.

## Navigating and using Microbe-ID

Navigating the demonstration site of Microbe-ID (http://Microbe-ID.org) the user will encounter a menu bar with links to Sequence-ID, Genotype-ID, and an about page. The home page has a general description of the functionality and components implemented in Microbe-ID, as well as links to the github site. Since input from the user is entered into forms as a query, each form implemented in Microbe-ID is encoded to check that the format of the data supplied by the user is supported. If the format is incorrect, the page will prompt the user with an error message, making the use of Microbe-ID more user-friendly.

## Sequence-ID

We created a module called Sequence-ID that uses BLAST analysis of common sequence loci for species identification (Fig. S1). Sequence-ID includes a PERL_CGI script that permits the communication between a user interface form (implemented in HTML5) and a BLAST database for the desired sequence data. The form recognizes the input in FASTA format and uses BLASTN to search for the most similar DNA sequences in the marker database. The PERL-CGI script receives the information of the HTML tabulated output from BLAST and displays a table of hits to the end user. Sequence-ID is customizable as the FASTA data file can readily be updated using the makeblastdb program of the BLAST suite. It can also similarly be implemented for BLASTP analysis of amino acid sequences.

In the Sequence-ID webpage, the user will find two main tabs: A ''Blast'' tab in which the web-app is contained, and a ''Help'' tab which contains a link to a ''Site Help'' prompt. This ''Site Help'' prompt shows an example of the FASTA sequence format recognized by Sequence-ID. The user can copy the FASTA sequence and paste it in the ''Blast'' tab text form to perform a BLAST search on the example query, or can download the database in FASTA format. After the BLAST is performed, the web page will provide a table including the BLAST search results for the query of interest using the BLAST alignment format.

## Genotype-ID

Genotype-ID is a web application designed to be user-friendly, and to facilitate the interaction with R using the set of tools listed in Table 1 (Fig. 1). To develop the web application, we used the R package 'shiny' (*RStudio Team, 2015*). Shiny facilitates the interactions between user, server, and R (Fig. 1). The shiny web framework relies on reactive programming, which allows dynamic deployment of traditional R scripts in response to data input through a website console whereby results generated in R are subsequently pushed to the end user. Thus, ordinary R packages, which otherwise require familiarity with the language, can be deployed behind a user-friendly interface. Genotype-ID interacts principally with the R packages 'poppr' (*Kamvar, Tabima & Grünwald, 2014*), 'adegenet' (*Jombart, 2008*), 'ape' (*Paradis, Claude & Strimmer, 2004*), and 'pegas' (*Paradis, 2010*) (Table 1).

Genotype-ID has three different modules, which are shown as tabs on the website, each specific to different molecular markers: SSR/microsatellite data (SSR/Microsatellite data), multilocus genotype sequence types (MLST data), and AFLP, RFLP, SNP and other binary datasets (Binary (AFLP) data). The user supplies a query via the web framework that is read into R and queried against a curated dataset. Genetic distances (Table 2) are calculated between the query and genotypes in the reference database, with relationship of the query to the database presented either as a UPGMA or neighbor-joining dendrogram with bootstrap support or a minimum spanning network. These methods reconstruct relationships of the query relative to the genotypes found in the curated database. Each module has its own customization scheme to analyze each particular molecular marker type. The SSR data tool calculates Bruvo's distance (*Bruvo et al., 2004*) to reconstruct the UPGMA or NJ dendrogram and minimum spanning networks. The MLST data tool permits the comparison of user-submitted gene sequences in FASTA format. The MLST data tool uses MAFFT (*Katoh et al., 2002*) to align each query sequence to the corresponding curated gene

**Table 1  Open source computational tools required to install and deploy Microbe-ID on a server.**

| Tool | Description | Source | References |
|---|---|---|---|
| Ape | R package for phylogenetic and evolutionary analysis | http://ape-package.ird.fr/ | Paradis, Claude & Strimmer (2004) |
| BLAST | Basic Local Alignment and Search Tool implemented as an algorithm for comparing DNA, RNA or protein query sequences against a reference database | ftp://ftp.ncbi.nlm.nih.gov/blast/ executables/blast+/LATEST/ | Altschul et al. (1990) |
| Adegenet | R package for multivariate analysis of genetic data | https://github.com/ thibautjombart/adegenet/ | Jombart (2008) |
| Bootstrap | A framework for developing responsive, mobile-first projects on the web | http://www.getbootstrap.com/ | – |
| Microbe-ID | Set of web-apps for identification of species, genotypes, and strains of any organism | https://github.com/grunwaldlab/ Microbe-ID | This paper. |
| MAFFT | Multiple sequence alignment algorithm to find homology between sequences using Fourier algorithms | http://mafft.cbrc.jp/alignment/ software/ | Katoh et al. (2002) |
| Pegas | R package for analysis of population genetic data | http://ape-package.ird.fr/pegas. html | Paradis (2010) |
| Poppr | R package for genetic analysis of populations with mixed reproduction | https://github.com/grunwaldlab/ poppr | Kamvar, Tabima & Grünwald (2014) and Kamvar, Brooks & Grünwald (2015) |
| Shiny | Interactive web application framework for R | http://shiny.rstudio.com/ | – |

**Notes.**

—, reference not available; see website provided for information.

database. It then concatenates all separate alignments, calculates the genetic distance for the alignment, and reconstructs distance dendrograms and minimum spanning networks. Lastly, the binary tool uses molecular markers such as AFLPs or RFLPs to reconstruct relationships. This tool uses binary data (coded as 1 and 0) and different genetic distances (Table 2) to reconstruct the UPGMA or NJ dendrogram and the minimum spanning network.

## The Microbe-ID implementation

Microbe-ID contains implementations of each of the modules for different custom databases: the SSR implementation uses nine diploid SSR loci for the oomycete species *P. ramorum* (Fig. S2); the MLST implementation uses eight multilocus sequence types for the bacterial species *Clavibacter michiganensis* subsp. *michiganensis*; and a binary implementation that uses 56 loci for AFLP data for the oomycete *Aphanomyces euteiches*. Each of the implementations of Genotype-ID contains collapsible instructions on how to format the query and a link to download a tabulated file with example queries formatted for use in Genotype-ID. The user can download the spreadsheet in order to edit, copy, and paste their custom queries into the "Data input" form. To make analyses that use a random seed repeatable, the user has the option to specify a seed number. In MLST-ID and Binary-ID, the user can select the genetic distance to be used in the analysis. After

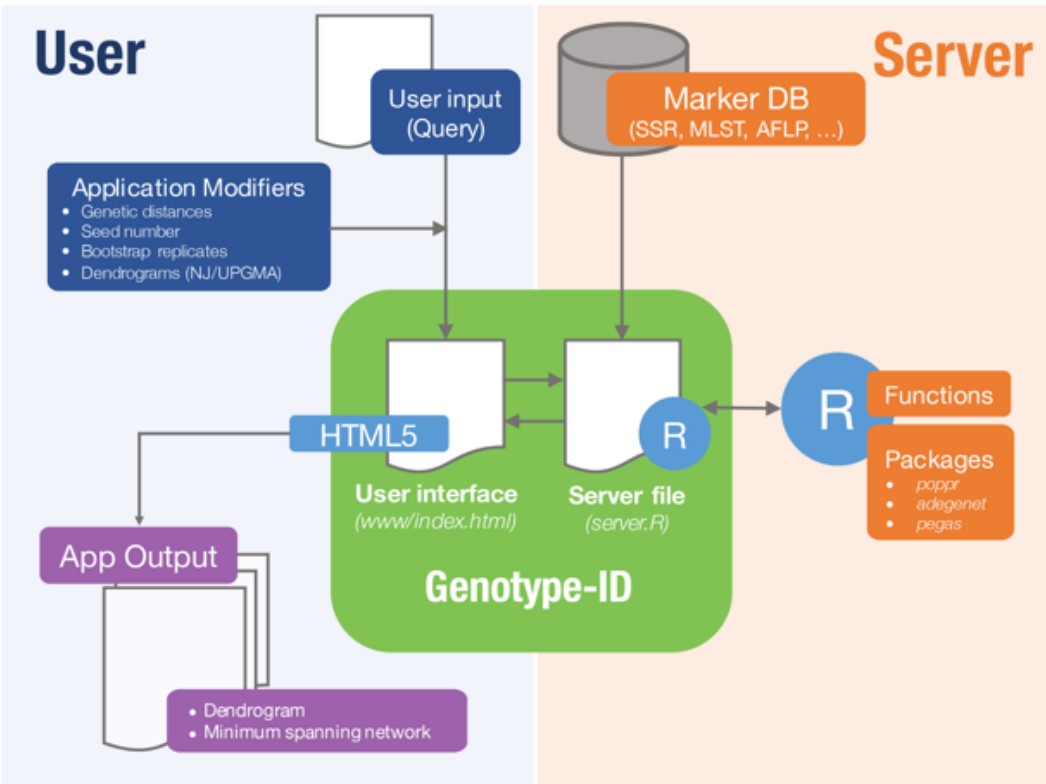

**Figure 1** **Diagram representing implementation of Genotype-ID, which is comprised of a user interface file (index.html) and a server file (server.R).** Each file communicates with the R framework (via shiny) and user (via HTML5). On the user side (left side), user input is provided by copy/paste of a query and selects/specifies the desired application modifiers (seed number, genetic distance calculation). This information is subsequently received and processed by the server file, prompting the application to run in R. On the server side (right side) a database file (Marker DB), R packages, and functions are retrieved and executed. When the run is complete, the server file provides output to the user interface file and displayed on the app output.

the user inputs the query and runs the web application, the web page will proceed to the *Analysis* section, where the user can choose between two different visualizations, a distance tree with bootstrap support values or a minimum spanning network. If the distance tree is selected, the user can change the tree algorithm (either neighbor joining or UPGMA) and number of bootstrap replicates. The user can also download results as a PDF or in NEWICK format. For the minimum spanning network, the user can adjust the grey scale for edge distances and download results as a PDF file. Implementations for other genetic data and different visualizations can be added.

### *Phytophthora*-ID implementation

Tools provided in Microbe-ID were implemented in http://Phytophthora-ID.org, a functional website for identifying samples from the genus *Phytophthora*, a group of economically important plant pathogens in the stramenopile branch of the tree of life (*Kamoun et al., 2015*). Phytophthora-ID version 1.0 (*Grünwald et al., 2011*) had a BLAST script to identify samples using the ITS barcode. *Phytophthora*-ID version 2.0

**Table 2 Genetic distances implemented in the Genotype-ID module of Microbe-ID.** Each of the distances included in Microbe-ID are specific to a given molecular marker used in the web application.

| Distance model | Module | R package | References |
|---|---|---|---|
| Felsenstein 81 (F81) | MLST-ID | ape | *Felsenstein (1981)* |
| Felsenstein 84 (F84) | MLST-ID | ape | *Felsenstein (1989)* |
| Indel | MLST-ID | ape | *Paradis, Claude & Strimmer (2004)* |
| Jukes-Cantor (JC69) | MLST-ID | ape | *Jukes & Cantor (1969)* |
| Kimura 80 (K80) | MLST-ID | ape | *Kimura (1980)* |
| Kimura 81 (K81) | MLST-ID | ape | *Kimura (1981)* |
| Raw | MLST-ID | ape | *Paradis, Claude & Strimmer (2004)* |
| Tamura and Nei 93 (TN93) | MLST-ID | ape | *Tamura & Nei (1993)* |
| Transitions (TS) | MLST-ID | ape | *Paradis, Claude & Strimmer (2004)* |
| Transversions (TV) | MLST-ID | ape | *Paradis, Claude & Strimmer (2004)* |
| Bruvo | SSR-ID | poppr | *Bruvo et al. (2004)* |
| Edwards | Binary-ID | poppr/adegenet | *Edwards (1971)* |
| Nei | Binary-ID | poppr/adegenet | *Nei (1972)* |
| Prevosti | Binary-ID | poppr/adegenet | *Prevosti, Ocaña & Alonso (1975)* |
| Reynolds | Binary-ID | poppr/adegenet | *Reynolds, Weir & Cockerham (1983)* |
| Rogers | Binary-ID | poppr/adegenet | *Rogers (1972)* |

**Notes.**
MLST-ID, Multilocus sequence typing; SSR-ID, SSR/microsatellite loci; Binary-ID, AFLP/SNP loci.

was substantially revised and upgraded from the first iteration and now includes a faster BLAST search implemented in Sequence-ID and a new genotype identification system implemented in Genotype-ID.

The current version of *Phytophthora*-ID contains a Sequence-ID module customized for two molecular barcodes used for *Phytophthora* species identification (ITS and *cox* spacer). This particular version of the sequence identification tool was created in PERL-CGI, and permits the search of a FASTA sequence query against a curated database of *Phytophthora* species. The PERL-CGI for *Phytophthora*-ID version 2.0 was also redesigned to run directly on the server to make the web application more stable and faster. In contrast, in *Phytophthora*-ID 1.0, the web application was designed as a communication wrapper to an external cluster, thus rendering functionality dependent on the external server.

We implemented the Sequence-ID module to use sequences from two genetic regions: the nuclear internal transcribed spacer (ITS) and mitochondrial *cox* spacer region spanning the *cox1* and *cox2* loci. Databases were created using sequences from published *Phytophthora* species descriptions (*Blair et al., 2008*; *Martin & Tooley, 2003*) or those that belong to classical *Phytophthora* species described at least 25 years ago and are still recognized as valid (*Erwin & Ribeiro, 1996*).

For the ITS region, we gathered a total of 211 sequences representing 108 species. For the *cox* spacer region, we created a database of 150 sequences representing 106 species. Laboratory protocols for preparing samples for sequence analysis are available on the website and in a previous publication (*Grünwald et al., 2011*). Additionally, a file with the complete set of ITS sequences, *cox* spacer sequences, GenBank accession numbers, and

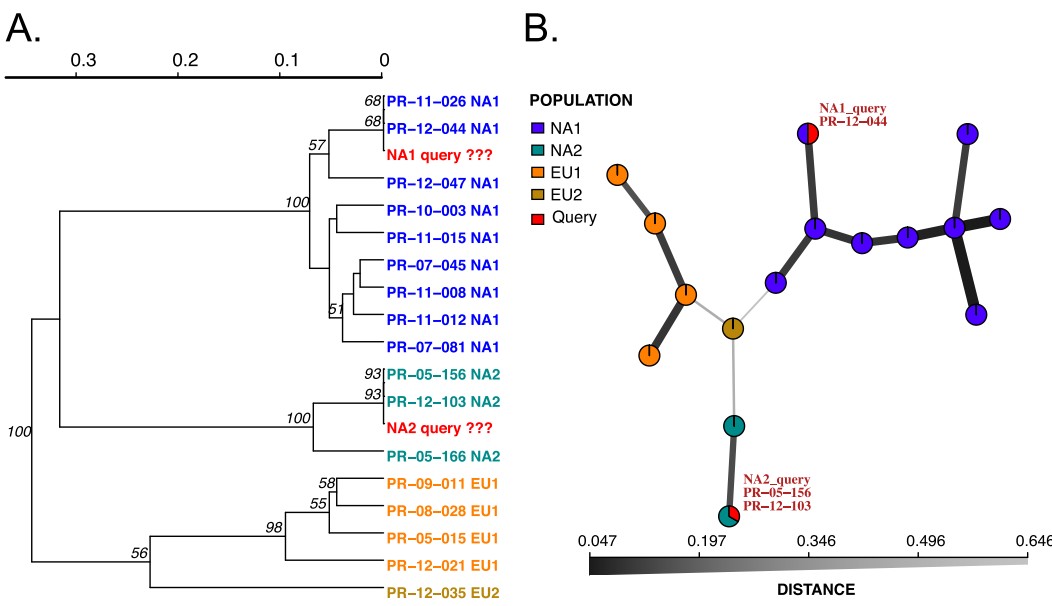

**Figure 2 Results of SSR-ID for NA1 and NA2 queries of *P. ramorum* provided in the example data file.** Each color represents a clonal lineage pre-assigned to each reference sample (NA1, NA2, EU1, EU2) with queries colored in red. (A) UPGMA tree with 1,000 bootstrap replicates and support values above branches. Queries are represented in red and all are correctly placed with reference samples of the presumptive clonal lineage while also representing the relationship between clonal lineages in the reference dataset. (B) Minimum spanning network reconstruction. Edge shade and width are inversely proportional to Bruvo's distance as shown in the horizontal scale bar. Queries are represented in red and placed in nodes with the most similar reference sample in the dataset, indicating the NA1 query is most similar to the PR-12-044 reference sample and the NA2 query is more closely related to the PR-05-156 and PR-12-103 samples, which also belong to the NA2 clonal lineage.

*Phytophthora* spp. can be downloaded from the website. Documentation on updates to the databases is provided on the website.

We implemented two modules for genotype identification of two *Phytophthora* species: *P. infestans* that causes potato late blight, and *P. ramorum* that causes sudden oak death. *P. infestans* has more than 18 reported clonal lineages (*Hu et al., 2012*), while *P. ramorum* has 4 reported clonal lineages (*Grünwald et al., 2009*; *Van Poucke et al., 2012*). To establish a database with a wide representation of clonal lineages in each species, we obtained and prepared DNA samples from *P. infestans* and *P. ramorum* that were collected from various regions of the world. A total of 48 *P. ramorum* isolates representing the 4 reported clonal lineages (NA1, NA2, EU1, EU2) were genotyped at nine SSR loci (*Prospero et al., 2007*; *Vercauteren et al., 2010*; *Grünwald et al., 2009*; *Ivors et al., 2006*). Similarly, 11 *P. infestans* clonal lineages that are dominant in the US (including US11, US12, US8, US20, US21, US23, US24, EU4, EU5, EU8, EU13) represented by a total population of 96 isolates were genotyped at 11 SSR loci, using protocols from *Lees et al. (2006)* and *Li et al. (2013)*. We constructed SSR reference databases compatible with the poppr/adegenet R packages that provide dendrograms and minimum spanning networks for queries. Bootstrap support can be calculated for dendrograms. To test the datasets, we used two queries per dataset, one sample of NA1 and NA2 clonal lineages each for *P. ramorum* and one sample of clonal

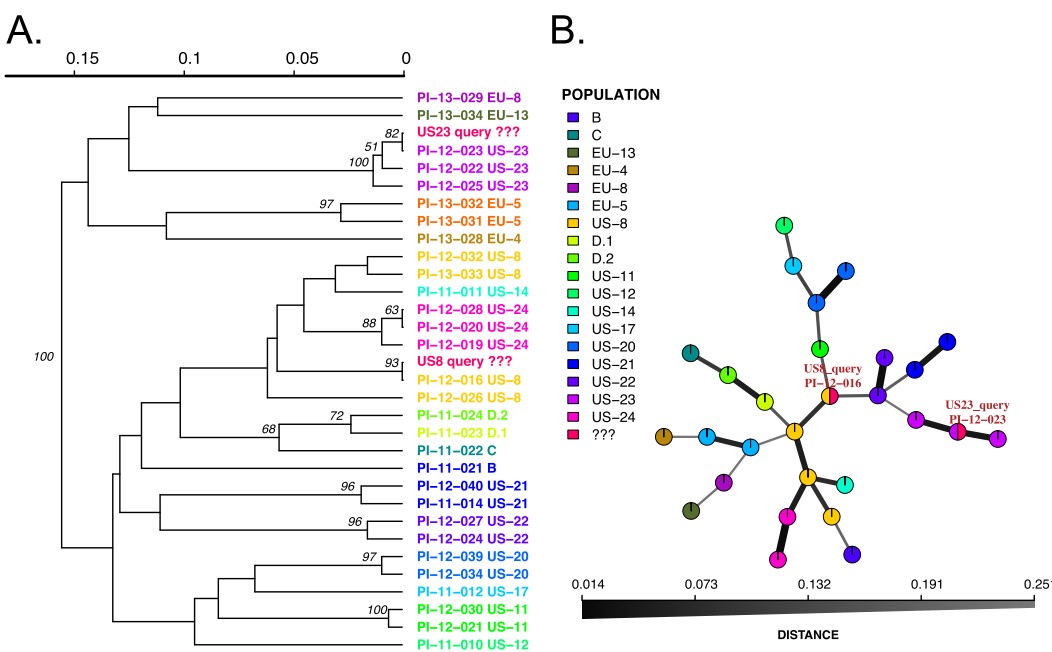

**Figure 3** **Results of SSR-ID queries for strains placed into the US8 and US23 clonal lineages of the potato late blight pathogen,** *P. infestans.* Colors correspond to clonal lineages assigned to each reference sample (B, C, EU-13, EU-14, etc.) except for the queries which are colored in red. (A) UPGMA tree with 1,000 bootstrap replicates with support values above branches. Queries are represented in red and all are correctly placed with samples of the presumptive clonal lineage while also representing relationships between clonal lineages in the reference dataset. (B) Minimum spanning network reconstruction. Edge shade and width are proportional to Bruvo's distance shown in the horizontal scale bar. Queries are represented in red nodes and appear in legend as '???'. Queries placed in nodes with the most similar reference sample, indicating that the US8 query is most similar to the PI-12-016 reference sample (US-8 clonal lineage) and the US23 query is most closely related to the PI-12-023 sample, part of the US-23 lineage.

lineages US-23 and US-8 clonal lineages for *P. infestans*. All samples grouped with the corresponding reference lineages with high support values (Figs. 2 and 3), demonstrating functionality of the Microbe-ID tool to identify samples correctly.

## CONCLUSIONS

We constructed a web framework that can use a wide array of molecular markers to rapidly determine identities of species and genotypes. This web framework is provided as open source code on GitHub (https://github.com/grunwaldlab/microbe-ID). We implemented a demonstration available at http://Microbe-ID.org. New implementations for any organisms require reference databases, scripts, and web pages including choices from the set of computational tools shown in Table 1. The Microbe-ID framework is currently implemented for two fully functional sites, *Phytophthora*-ID (http://Phytophthora-ID.org/) and Gall-ID (http://Gall-ID.cgrb.oregonstate.edu/; *Davis II et al., 2016*).

Modularity of Microbe-ID permits implementation of a range of markers using curated databases of known species or genotypes to determine identity of specimens. Use of a custom BLAST database permits identification of any sample of a species of interest by using any sequence-based molecular marker. Note that the amount of time required for

BLAST execution is dependent on size of the databases. For the Sequence-ID ITS region implementation for *Phytophthora*, each run was completed and loaded to the web-app in less than one second, with results shown as a BLAST output table.

Most innovative in this web framework is the Genotype-ID module, which permits the researcher to use, in a simple, web-based interface, any type of molecular marker with a corresponding and curated reference dataset. Use of a shiny server for strain identification is completely novel and modularity permits use of virtually any molecular marker suitable for existing R packages. For genotyping, population genetic markers demonstrated include SSR, MLST, and AFLP/RFLP data, but can also be expanded to presence/absence of genes or alleles or any other genetic features of interest. Custom implementations must include a reference dataset and custom R scripts. Implementations can be customized for virtually any organism and any molecular marker that can be analyzed in R.

Development of tools for species identification has increased in recent years; but lack of modularity and complicated methodological procedures make the process of species identification tedious and challenging for researchers without sufficient skills in computational biology. We developed an easy-to-set-up web framework, which permits quick and flexible deployment of species identification and genotyping tools using sequence, microsatellite/SSR, AFLP/RFLP, or MLST data for any organism. Given use of the bootstrap html framework, webpages are light in terms of computer resources required and are thus usable on any device including smartphones and tablets, and function with any browser. These tools have been deployed and used successfully for the genus *Phytophthora* and gall-forming bacteria, providing examples of two curated websites (*Grünwald et al., 2011*; *Davis II et al., 2016*).

### Funding
This research is supported in part by the US Department of Agriculture (USDA) Agricultural Research Service Grant 5358-22000-039-00D (NJG), USDA National Institute of Food and Agriculture (NIFA) Grant 2011-68004-30154 (NJG), the USDA ARS Floriculture Nursery Research Initiative (NJG), USDA NIFA Grant 2014-51181-22384 (JHC and NJG), and USDA NIFA 2012-67012-19844 (SEE). The funders had no role in study design, data collection and analysis, decision to publish, or preparation of the manuscript.

### Grant Disclosures
The following grant information was disclosed by the authors:
US Department of Agriculture (USDA) Agricultural Research Service: 5358-22000-039-00D.
USDA National Institute of Food and Agriculture (NIFA): 2011-68004-30154.
USDA ARS Floriculture Nursery Research Initiative.
USDA NIFA: 2014-51181-22384, 2012-67012-19844.

### Competing Interests
Niklaus Grünwald and Jeff Chang are Academic Editors for PeerJ.

## Author Contributions

- Javier F. Tabima conceived and designed the experiments, performed the experiments, analyzed the data, contributed reagents/materials/analysis tools, wrote the paper, prepared figures and/or tables, reviewed drafts of the paper.
- Sydney E. Everhart conceived and designed the experiments, analyzed the data, wrote the paper, reviewed drafts of the paper.
- Meredith M. Larsen and Matthew A. Tancos performed the experiments, contributed reagents/materials/analysis tools, reviewed drafts of the paper.
- Alexandra J. Weisberg and Zhian N. Kamvar performed the experiments, analyzed the data, contributed reagents/materials/analysis tools, reviewed drafts of the paper.
- Christine D. Smart and Jeff H. Chang wrote the paper, reviewed drafts of the paper.
- Niklaus J. Grünwald conceived and designed the experiments, wrote the paper, reviewed drafts of the paper.

## Data Availability

https://github.com/grunwaldlab/Microbe-ID.

## Supplemental Information

Supplemental information for this article can be found online at http://dx.doi.org/10.7717/peerj.2279#supplemental-information.

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
