# Peer review of "Microbe-ID: an open source toolbox for microbial genotyping and species identification"

_PeerJ, doi:10.7717/peerj.2279_

## Round 0.1 · original submission · Minor Revisions

· Academic Editor

Minor Revisions

I found your manuscript very interesting and clear for the possibilities that offer this web-based set of applications. However, there are a few comments raised by the reviewers that need clarification to improve the clarity of the manuscript. Please explain how the web platform will be maintained and implemented adding new options and analysis and the possibility for scientific community to continue upgrading it via GitHub

Reviewer 1 ·

Basic reporting

No Comments

Experimental design

No Comments

Validity of the findings

No Comments

Additional comments

In my opinion there are general questions that could be explained in the manuscript such as the quality of the sequences used for the toolbox. Also, are they based on the type specimens? Is there a curator for the database? Are there enough funds to keep it running? Is the data obtained using the toolbox good for publication purposes?

Reviewer 2 ·

Basic reporting

No comments

Experimental design

No comments

Validity of the findings

No comments

Additional comments

Dear authors,
in your paper you present a new web framework for the identification of species and genotypes. This second part of your framework is particularly innovative and interesting. I am personally convinced that such a tool will be useful for numerous scientists. The manuscript is well written and clear and I only have a very few minor comments. Therefore, I recommend to accept this paper after minor revision.

Specific comments
ll. 51-52: I don’t exactly understand the meaning of this sentence. Can you please rewrite it?
l. 65: Etter et al. 2011, in the references 2010. Please verify
l. 144: is Rstudio an official reference for the package “shiny”? When yes, then it should be in the references.
Fig. 1 and Fig. 3: in my opinion, not really necessary

Reviewer 3 ·

Basic reporting

no comments

Experimental design

no comments - this is reporting database development not an experiment

Validity of the findings

no comments

Additional comments

I am somewhat confused by this manuscript which could mean that i have not read it correctly. The introduction starts with 'Development of tools to identify species, genotypes, or novel strains of invasive organisms is critical for monitoring emergence and implementing rapid response measures. Molecular markers, although critical to identifying species or genotypes, require bioinformatic tools for analysis. However, user-friendly analytical tools for fast identification are not readily available. To address this need, we created a web-based set of applications called Microbe-ID that allow for customizing a toolbox for rapid species identification and strain genotyping using any genetic markers of choice.' To me this sounds like a big database full of lots of information for many invasive organisms - but as i continued to read it is my understanding that the database presented here only include data for oomycetes, predominantly Phytophthora. In fact it jus seems to be an update of Phytophthora-ID including some genotypying information for a small group of species. The database does not even include all currently described Phytophthora species.

I may have completely missed the point and i tried to go to the MICROBE-ID website to check it out, but it is not live

If this database is in fact only for Phytophthora/oomycetes - i think this should be very clear from the start rather than all the background on other organisms. Then you could just conclude that other people could submit their datasets to grow the database.

otherwise I have no comments about the structure or the flow of the manuscript - it is well written and well illustrated.

Reviewer 4 ·

Basic reporting

1. Article is well written and easy to understand.
2. Introductory material was easy to understand.
3. Structure (comment below) was a bit odd for a scientific manuscript

Experimental design

1. This manuscript is more computational in nature and doesn't really have a primary research question. All that said I think it is still important information for the broader medical community.
2. Good supporting details all included and external links worked for me.
3. I couldn't find the captions for the supplemental figures.

Validity of the findings

1. Data was robust and sound
2. Research was original but builds very significantly on Phytophthora-ID and there is some overlap.
3. Minimal speculation, resources developed a well cited. I am a bit concerned that none of the materials cited will actually go with the manuscript. So if the Microbe-ID server ever went down this manuscript would be irrelevant (as have other papers of this type)

Additional comments

This manuscript entitled "Microbe-ID: An open source toolbox for microbial genotyping and species identification” discusses species and genotyping software tools for identification of various microbes. The PIs focused these tools on the genus Phytophthora utilizing various genotyping techniques like SSR analysis. A nice component of this manuscript is the authors are trying to show molecular tools in Plant Pathology and make them available to other research communities. The manuscript is appropriate for publication in PeerJ and I recommend publication once the authors address the following comments.

Main points
1. Part of me feels like the title isn’t descriptive enough but I realize this is primarily a BioMed journal. It is just there is no little mention of Phytophthora in the title and all of the research shown is about Phytophthora. Perhaps: "Adapting database tools to develop Microbe-ID, an open source toolbox for genus and species identification"
2. The manuscript is very well written, but the overall format seems a bit odd, and contains a background section with no results, methods or discussion and is basically just lumped into one large section. This format might be alright for this type of manuscript (mainly software) but seems a bit different. I am comfortable deferring to the editor on this matter.
3. I am a bit concerned about the novelty of “species-identification” portion of this manuscript since Phytophthora-ID has been around for several years (reported in 2011).
4. I feel like what makes this manuscript is mainly going on in the background. This paper is more of an announcement of the technology. My suggestion would be to provide a bit more of a protocol embedded in the manuscript on how a user would set up their own database. The first question I have when I see this manuscript is how will I adapt this tool to my system (especially since it is appealing to the medical community) or am I better off using tools such as BioEdit.
5. For additional loci (atp9, nad9, rps10 etc) that are important for Phytophthora I would add these recent manuscripts (Martin et al., 2014 Fungal Genet Biol.) somewhere around line 200.
6. Tables 1 and 2. Add longer captions, in general they should be able to stand alone.
7. In Figure 2. I would remove “Etc." just because it seems irrelevant
8. I am not sure if Figures 1 and 3 are really necessary. They seem fairly large in the proof.

---

## Round 0.2 · accepted · Accept

· Academic Editor

Accept

I think that you satisfactorily addressed all comments raised by the reviewers and thank you for your propmot response. I believe that the Microbe-ID toolbox will be very useful for the scientific community